# Prediction of the Medicinal Mechanisms of *Pinellia ternata Breitenbach*, a Traditional Medicine for Gastrointestinal Motility Disorders, through Network Pharmacology

**DOI:** 10.3390/plants11101348

**Published:** 2022-05-19

**Authors:** Na Ri Choi, Jongwon Park, Seok-Jae Ko, Jeong Nam Kim, Woogyun Choi, Jae-Woo Park, Byung Joo Kim

**Affiliations:** 1Division of Longevity and Biofunctional Medicine, School of Korean Medicine, Pusan National University, Yangsan 50612, Korea; nariring@gmail.com (N.R.C.); dasom721@naver.com (J.N.K.); ak0510@hanmail.net (W.C.); 2Department of Clinical Korean Medicine, Graduate School of Kyung Hee University, Seoul 02447, Korea; 7jwpark@daum.net (J.P.); kokokoko119@daum.net (S.-J.K.); 3Department of Gastroenterology, College of Korean Medicine, Kyung Hee University, Seoul 02447, Korea

**Keywords:** *Pinellia ternata Breitenbach*, gastrointestinal motility disorders, network-based systems pharmacological, traditional medicine

## Abstract

*Pinellia ternata Breitenbach* (PTB) is a widely used herbal medicine in China, Japan, and South Korea. It has antiemetic, anti-inflammatory, antitussive, and sedative properties. The raw material is toxic, but can be made safer using alum solution or by boiling it for a long time. In addition, PTB seems to be effective for gastrointestinal motility disorders (GMDs), but this is yet to be conclusively proven. Herein, PTB compounds, targets, and related diseases were investigated using the traditional Chinese medical systems pharmacology database and an analysis platform. Information on target genes was confirmed using the UniProt database. Using Cytoscape 3.8.2, a network was established and GMD-related genes were searched using the Cytoscape stringApp. The effects of the PTB extract on the pacemaker potential of interstitial cells of Cajal and GMD mouse models were investigated. In total, 12 compounds were found to target 13 GMD-related genes. In animal experiments, PTB was found to better regulate pacemaker potential in vitro and inhibit GMD signs compared to control groups in vivo. Animal studies showed that the mechanism underlying the effects of PTB is closely related to gastrointestinal motility. The results obtained demonstrated that PTB offers a potential means to treat GMDs, and we suggested that the medicinal mechanism of GMDs can be explained by the relationship between 12 major components of PTB, including oleic acid, and 13 GMD-related genes.

## 1. Introduction

Gastrointestinal motility disorders (GMDs) can occur anywhere in the gastrointestinal tract. Therefore, such conditions can exhibit a variety of chronic symptoms that significantly affect a patient’s quality of life, including nausea and vomiting [1,2,3]. Currently, there is no cure for GMD, and management may involve lifestyle changes and drugs. The gastrointestinal process occurs in each segment of this organ, and the contents move between each segment through active and passive peristaltic movements, which are slow waves of muscle contraction and relaxation [4,5]. Smooth muscle cells, intestinal neurons, including telosites, and interstitial cells of Cajal (ICCs) play an important role in gastrointestinal motility. Various studies have shown that ICCs act as special gastrointestinal pacemaker cells [6,7]. Thus, abnormalities in ICCs are related to several GMDs such as gastroparesis [8].

*Pinellia ternata Breitenbach* (PTB) is one of the main ingredients in traditional herbal medicine and has been used for antiemetic, antitussive, anti-inflammatory, and sedative purposes since ancient times [9]. Previously characterized phytochemicals in this herb include alkaloids [10], volatile oils [11], and polysaccharides [12]. To investigate the effects of PTB on GMDs and the medicinal mechanisms, we used a network-based system pharmacological approach. We used the traditional Chinese medicine systems pharmacology database and analysis platform (TCMSP, https://tcmsp-e.com/tcmsp.php. accessed date: 10 January 2022 to 15 March 2022), which provides comprehensive information on the relationship between herbs and their components, molecular targets, and diseases. TCMSP provides information on the absorption, distribution, metabolism, and excretion (ADME), which is useful for developing new drugs or analyzing the efficacy of herbal medicines [13]. A schematic of the research protocol is shown in Figure 1. In this study, a network-based pharmacological analysis of PTB was performed. In addition, PTB extracts were examined for their pacemaking activity and evaluated for their therapeutic effects in mouse models of GMDs.

## 2. Results

### 2.1. Information for 366 Targets Derived through Correlation Investigation between Compounds and Targets

We identified 116 potentially active compounds in PTB using the TCMSP database (Appendix A). Among these, 85 compounds contained the target information (Appendix A), and it was found that these 85 compounds and 366 targets interacted with each other through a combination of 1226 compounds (Figure 2). As shown in Figure 2, GLY was linked to the most target genes (161 genes), followed by DAL (65 genes), succinic acid (57 genes), L-serine (54 genes), ASI (47 genes), oleic acid (43 genes), beta-sitosterol (38 genes), baicalein (36 genes), D-2-aminobutyrate (34 genes), threonine (30 genes), stigmasterol (30 genes), and glutamine (29 genes).

### 2.2. Twenty Active Compounds Met the Criteria for ADME Parameters

Twenty compounds were included in the active compound screening criteria (Table 1) as follows: EIC, (3S,6S)-3-(benzyl)-6-(4-hydroxybenzyl)piperazine-2,5-quinone, 10,13-eicosadienoic acid, 12,13-epoxy-9-hydroxynonadeca-7,10-dienoic acid, 24-ethylcholest-4-en-3-one, 6-shogaol, 8-octadecenoic acid, baicalein, beta-sitosterol, cavidine, coniferin, cyclo-(leu-tyr), cyclo-(val-tyr), cycloartenol, gondoic acid, linolenic acid, methyl palmitelaidate, oleic acid, pedatisectine a, and stigmasterol.

### 2.3. Identification of 53 Compounds Related to Gastrointestinal (GI) Disease in PTB

We also investigated the compound–target–disease relationship using the TCMSP database. We found that 53 compounds were associated with GI diseases (Table 2). In particular, EIC, 6-shogaol, cavidine, baicalein, beta-sitosterol, methyl palmitelaidate, (3S,6S)-3-(benzyl)-6-(4-hydroxybenzyl)piperazine-2,5-quinone, linolenic acid, stigmasterol, coniferin, oleic acid, pedatisectine a, and cyclo-(val-tyr) were found to be active compounds associated with GI, and other compounds, including 3,4,5-trihydroxybenzoic acid, ANN, ASI, beta-D-ribofuranoside, xanthine-9, beta-elemene, caffeic acid, cis-p-coumarate, crysophanol, D-2-aminobutyrate, DAL, DTY, DUR, eciphin, ferulic acid (CIS), GLY, gamma-aminobutyric acid, glutamine, OMD, gynesine, HMF, hydroquinone, isolariciresino, istidina, L-arginine, leucinum, L-Ile, L-valine, L-pseudoephedrine, norharman, palmitic acid, pedatisectine f, pentadecylic acid, protocatechuic acid, sitogluside, stearic acid, succinic acid, threonine, THM, vanillic acid, and zoomaric acid, associated with gastrointestinal diseases, were classified as not active (Figure 3).

### 2.4. All 52 GI Disease-Related Compounds, Except 6-Shogaolin, in PTB Are Related to GMDs

To determine the relationship between PTB and GMDs, we used the Cytoscaping app to check for GMD-related genetic information. First, 100 GMD-related genes were identified by applying a reliability (score) cutoff of 0.40 and a maximum of 100 proteins (Appendix A). From the results obtained, a network of GMD-related genes and PTB target genes was created (Figure 4). It was confirmed that there were 13 genes corresponding to both sets of genes. The GMD-related genes targeted by PTB were actin, cytoplasmic 1 (*ACTB*), cholecystokinin (*CCK*), C-reactive protein (*CRP*), proto-oncogene c-Fos (*FOS*), glucagon (*GCG*), insulin (*INS*), myeloperoxidase (*MPO*), Mu-type opioid receptor (*OPRM1*), prostaglandin G/H synthase 2 (*PTGS2*), peptide YY (PYY), sodium channel protein type 5 subunit alpha (*SCN5A*), sodium-dependent serotonin transporter (*SLC6A4*), and transient receptor potential cation channel subfamily V member 1 (*TRPV1*).

### 2.5. The Network of GMD-Related Genes and Compounds for Identifying Molecules of Interest

Figure 5 illustrates the network of relationships between PTB compounds and GMD-related target genes. The results showed that oleic acid and PTGS2 are most closely related to GMD. In summary, EIC, 6-shogaol, cavidine, baicalein, beta-sitosterol, methyl palmitelaidate, (3S,6S)-3-(benzyl)-6-(4-hydroxybenzyl)piperazine-2,5-quinone, linolenic acid, stigmasterol, coniferin, oleic acid, pedatisectine a, and cyclo-(val-tyr) were determined to be active compounds that target GMD-related genes, indicating that these compounds might be potential medicinal candidates. However, as shown in Figure 2, PTB contained many compounds with multi-targeting characteristics; thus, the synergistic effects of PTB compounds on GMD were investigated in vitro and in vivo.

### 2.6. Effects of PTB Extract on the Pacemaker Potential of ICCs

ICCs regulate gastrointestinal motility by acting as pacemakers in the GI tract [6,7,8]. The whole-cell technique showed that ICCs spontaneously induced pacemaker potentials with an average resting membrane potential of −56.8 ± 1.6 mV and amplitude of 24.9 ± 1.2 mV (Figure 6). PTB extract (10–300 μg/mL) depolarized pacemaker potentials and decreased the amplitude (Figure 6A–D). The average depolarization was 6.9 ± 0.9 mV (*p* < 0.0001) at 10 μg/mL, 11.2 ± 1.0 mV (*p* < 0.0001) at 50 μg/mL, 24.8 ± 1.2 mV (*p* < 0.0001) at 100 μg/mL, and 31.8 ± 1.1 mV (*p* < 0.0001) at 300 μg/mL (Figure 6E). The average amplitude was 24.2 ± 0.8 mV at 10 μg/mL, 9.8 ± 1.2 mV (*p* < 0.0001) at 50 μg/mL, 2.9 ± 1.0 mV (*p* < 0.0001) at 100 μg/mL, and 1.5 ± 0.5 mV (*p* < 0.0001) at 300 μg/mL (Figure 6F). These results indicated that PTB extract regulates the pacemaker potential of ICCs.

### 2.7. Importance of Ca^2+^ in PTB Extract-Induced Pacemaker Potential Depolarization of ICCs

To investigate the importance of Ca^2+^ in PTB extract-induced responses, we used external Ca^2+^-free conditions or thapsigargin. Pre-treatment with an external Ca^2+^-free solution or thapsigargin suppressed pacemaker potentials and inhibited PTB extract-induced responses (Figure 7A,B). The average depolarization was 1.1 ± 0.5 mV (*p* < 0.0001) with Ca^2+^-free solution and 1.2 ± 0.3 mV (*p* < 0.0001) with thapsigargin (Figure 7C). However, the average amplitude did not change significantly (Figure 7D). These results indicated that the PTB extract-induced responses are controlled by Ca^2+^.

### 2.8. Importance of Muscarinic M3, 5-HT3, and 5-HT7 Receptors in PTB Extract-Induced Pacemaker Potential Depolarization of ICCs

Muscarinic and 5-HT receptors are expressed in the GI tract and associated with gastrointestinal motility [14,15]. Murine small intestinal ICCs express muscarinic M2, M3, or 5-HT3,4,7 receptors [16,17]. Pre-treatment with 4-DAMP (a muscarinic M3 receptor antagonist) inhibited the PTB-induced effects (Figure 8A). However, methoctramine, (a muscarinic M2 receptor antagonist) had no effect (Figure 8B). In addition, Y25130 (a 5-HT3 antagonist) and SB269970 (a 5-HT7 antagonist) inhibited PTB-induced effects (Figure 8C,E). However, RS39604 (a 5-HT4 antagonist) showed no effect (Figure 8D). The average depolarization was 6.9 ± 0.8 mV (*p* < 0.0001) with 4-DAMP, 25.6 ± 1.2 mV with methoctramine, 6.1 ± 0.7 mV (*p* < 0.0001) with Y25130, 23.6 ± 1.2 mV with RS39604, and 6.0 ± 0.9 mV (*p* < 0.0001) with SB269970 (Figure 8F). The average amplitude was 13.5 ± 0.7 mV (*p* < 0.0001) with 4-DAMP, 3.5 ± 1.1 mV with methoctramine, 14.2 ± 0.4 mV (*p* < 0.0001) with Y25130, 4.2 ± 0.9 mV with RS39604, and 14.2 ± 0.8 mV (*p* < 0.0001) with SB269970 (Figure 8G). These results suggested that PTB affects pacemaker potentials via the M3, 5-HT3, and 5-HT7 receptors.

### 2.9. Effects of PTB Extract on the ITR

The ITR was 51.0 ± 3.2% (Figure 5). The PTB extract increased the ITR to 50.1 ± 3.2% at 0.01 g/kg, 58.9 ± 6.3% at 0.1 g/kg, and 76.1 ± 6.1% (*p* < 0.0001) at 1 g/kg (Figure 9A). In addition, AA decreased ITR (32.5 ± 4.2% vs. 50.3 ± 2.6% in normal mice; P < 0.0001; Figure 9B). However, PTB extract restored this response to 52.5 ± 1.7% (*p* < 0.0001), 51.3 ± 3.8% (*p* < 0.0001), and 56.8 ± 1.9% (*p* < 0.0001), respectively (Figure 9B). These results indicated that the PTB extract can increase the normal ITR and recover the ITR in GMD mice.

## 3. Discussion

PTB is a monocotyledonous perennial herbaceous plant that is widely used in traditional herbal medicine in China, Japan, and South Korea. It has been used for antiemetic, anti-inflammatory, antitussive, and sedative purposes [18]. In addition, experiments using a mouse model have shown that PTB induces efferent activity in the gastric branches of the vagus nerve, which is known to be effective in GMDs, but the pharmacological mechanisms underlying this have yet to be studied [19]. Therefore, this study was conducted using a combination of network-based pharmacological analyses and experimental verification to identify the physiologically active ingredients and medicinal mechanisms of PTB. Additionally, 116 compounds were identified in PTB, including 19 active compounds (Sup. 1). Of these 116 compounds, 85 had target information and 365 target genes were collected (Sup. 2). Twelve genes were associated with GMD (Figure 5), including *ACTB*, *CCK*, *CRP*), *FOS*, *GCG*, *INS*, *MPO*, *OPRM1*, *PTGS2*, *PYY*, *SCN5A*, *SLC6A4*, and *TRPV1* (Figure 4). These results are the same as those of past studies. Specifically, as shown in Table 2 and Figure 5, *PTGS2* was the target of all GMD-related PTB compounds, suggesting that PTB compounds can adjust *PTGS2* levels synergistically. PTGS2 plays the most important role in the treatment of mucosal defense and gastrointestinal inflammation and ulcers. PTGS2 also contributes to normalization of gastrointestinal function after inflammation [20]. In addition, SCN5A and OPRM1 were found to be targets of multiple PTB compounds. The voltage-gated mechanosensitive Na^+^ channel NaV1.5, encoded by *SCN5A*, is present in ICCs and human gastrointestinal smooth muscle cells. SCN5A contributes to the electrical slow wave and mechanical sensitivity of smooth muscles [21]. OPRM1 is linked to the inhibition of acetylcholine release from intestinal and motor neurons and the inhibition of purine and nitrogen release from motor neurons, thereby inhibiting propulsion kinetic patterns [22]. OPRM1 activation induces the inhibition of submucosal secretion by motor neurons, which reduces active Cl secretion and passive liquid movement to the colon lumen [23]. This effect on mobility and secretion results in constipation induced by OPRM1. These results indicate that the medicinal mechanism of GMDs is related to the effects of PTB on PTGS2, SCN5A, and ORPM1. The GMD-related active compounds EIC, (3S,6S)-3-(benzyl)-6-(4-hydroxybenzyl)piperazine-2,5-quinone, baicalein, beta-sitosterol, cavidine, coniferin, cyclo-(val-tyr), linolenic acid, methyl palmitelaidate, oleic acid, pedatisectine a, and stigmasterol were identified (Figure 5). In addition, 12 compounds were found to target PTGS2, and oleic acid was found to target PTGS2, MPO, CCK, CRP, GCG, INS, and PYY. Beta-sitosterol targets PTGS2, SCN5A, OPRM1, and SLC6A4, and cavidine targets PTGS2, SCN5A, OPRM1, and SLC6A4. Several studies have reported a relationship between the major components of PTB and GMDs. Emulsions containing oleic acid activate nutrient-induced inhibitory feedback mechanisms in the small intestine, slowing down gastrointestinal passage and reducing diarrhea [24]. beta-sitosterol has significant antibacterial activity and improved DSS-induced colitis in mice [25]. Cavidine improved ulcerative colitis by regulating the oxidation and antioxidant balance and inhibiting NF-κB signaling pathways and pro-inflammatory cytokines, such as TNF-α and IL-6, in colonic tissue [26]. As shown in Figure 2, compounds interacted with an average of 14 target genes upon identifying the multi-compound multi-target properties of herbal medicines, and PTB was predicted to be a therapeutic medicine for GMDs based on the synergies among several compounds contained in PTB.

We investigated the restorative effects of PTB in a mouse model of GMDs. First, we checked the effects of PTB extract on the pacemaker potentials of ICCs. PTB extract was found to regulate the pacemaker potential of ICCs, and this response was controlled by Ca^2+^ (Figure 6 and Figure 7). In addition, PTB reactions occurred through M3, 5-HT3, and 5-HT7 receptors (Figure 8). Further, PTB extract increased the normal ITR and restored the ITR in GMD mice (Figure 9). In summary, our results showed that PTB has potential for the treatment of GMDs, and that its medicinal effect may be due to the regulation of ICCs. They also suggested that these mechanisms are related to the interactions among 12 key components of PTB, such as oleic acid, and 13 GMD-related genes, including *PTGS2*.

## 4. Materials and Methods

### 4.1. Network-Based Pharmacological Analysis by PTB

#### 4.1.1. Identification of PTB Compounds

Analysis platforms were used to identify the potential active compounds of PTB. We entered ‘*Pinellia ternata Breitenbach*’ as a query to search for the herb name.

#### 4.1.2. Analysis of Targets

The target information of the compound was determined by searching TCMSP [13]. The target proteins were linked to the official gene names using the UniProtKB database (https://www.uniprot.org/uniprot. accessed date: 10 January 2022 to 15 March 2022) [27].

#### 4.1.3. Network Analysis

The compound–target network was constructed using Cytoscape 3.8.2 (https://cytoscape.org. accessed date: 10 January 2022 to 15 March 2022) [28]. GMD-related genes were collected using the Cytoscaping app [29].

#### 4.1.4. Active Compound Screening

Using ADME parameters such as molecular weight, oral bioavailability (OB), Caco-2 permeability (Caco-2), and drug likeness (DL), physiologically active compounds in PTB were screened using the following criteria: OB ≥ 30%, DL ≥ 0.10, and Caco −2 ≥ −0.4. Compounds with values that met the criteria applied were considered active compounds.

### 4.2. Animal Experiments

#### 4.2.1. Preparation of the PTB Extract

Dried PTB was purchased from Herb Farm Co. (Wonju, Korea). A voucher specimen (PJW-058) was deposited at the Department of Herbology, College of Korean Medicine, Kyung Hee University, Korea. Dried PTB (400 g) was extracted using distilled water (4000 mL) for 2 h at 100 °C. The water extract of PTB was passed through a membrane filter (0.45 μm; EMD Millipore). After evaporation, the remaining aqueous extract was freeze-dried to yield 8.5% of the dried weight (*w*/*w*).

#### 4.2.2. Preparation of ICCs

A total of 74 ICR mice (37 males and 37 females; 3–8 days-old) were used for the experiments. Small intestinal cells were cultured with smooth muscle growth medium (Clonetics, San Diego, CA, USA).

#### 4.2.3. Electrophysiological Experiments

KCl 5 mM, NaCl 135 mM, CaCl2 2 mM, glucose 10 mM, MgCl2 1.2 mM, and HEPES 10 mM were used as bath solutions. KCl 140 mM, MgCl2 5 mM, K2ATP 2.7 mM, NaGTP 0.1 mM, creatine phosphate disodium 2.5 mM, HEPES 5 mM, and EGTA 0.1 mM were used as pipette solutions. Whole-cell configuration was used.

#### 4.2.4. Intestinal Transit Rate (ITR)

A total of 45 ICR mice (males, 5–6 weeks-old) were used for ITR experiments. Evans blue (5%, w/v) was administered after the administration of PTB extract into the stomach. After 30 min of Evans blue administration, the ITR was measured.

#### 4.2.5. GMD Model Mice

Acetic acid (AA, 0.6%, w/v, in saline)-induced peritoneal stimulation was used to generate a GMD mouse model. AA was injected intraperitoneally, and other processes were the same as previous studies [30,31].

#### 4.2.6. Statistical Analyses

Data are expressed as the mean ± standard error of the mean. Significant differences were evaluated using one-way analysis of variance or a student’s t-test. Statistical significance was considered as *p* < 0.05.

## 5. Conclusions

PTB analysis using a network-based pharmacological approach showed that 12 compounds and 13 genes were associated with GMDs. Our animal studies showed that PTB regulates the pacemaker potential of ICCs and inhibits GMD-like signs in a mouse model of GMDs. These results indicate that PTB has therapeutic potential for GMD treatment. In addition, we proposed a mechanism responsible for the interactions among 12 PTB compounds and 13 GMD-related genes.

## Figures and Tables

**Figure 1 plants-11-01348-f001:**
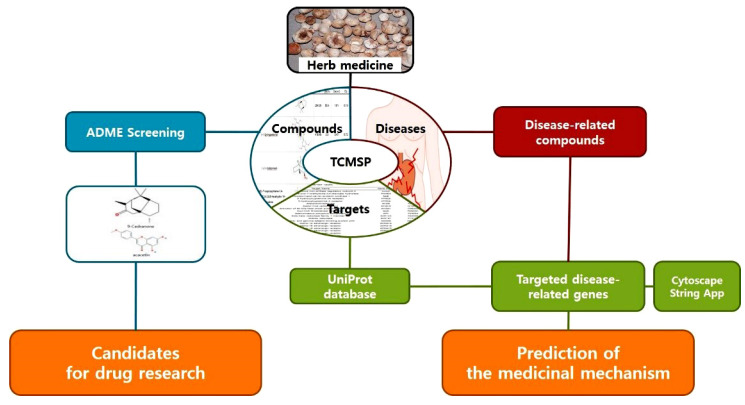
Schematic of the study protocol for network pharmacology exploration. ADME: absorption, distribution, metabolism, and excretion.

**Figure 2 plants-11-01348-f002:**
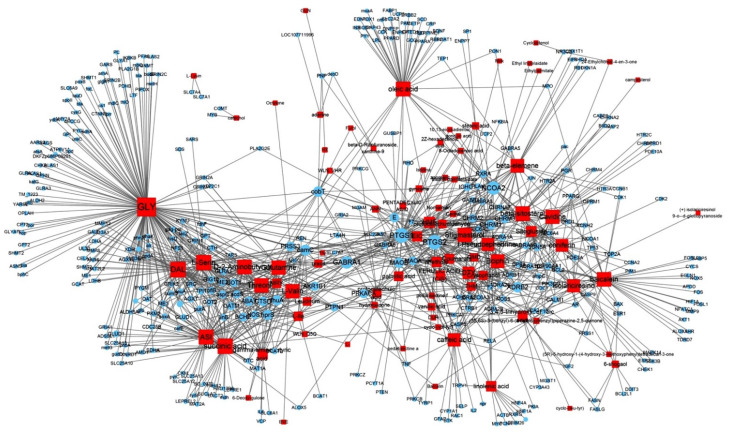
Compound–target network of *Pinellia ternate Breitenbach*. The size of the node depends on the number of connected edges. The compounds are expressed as red square nodes, and the targets are expressed as blue round nodes.

**Figure 3 plants-11-01348-f003:**
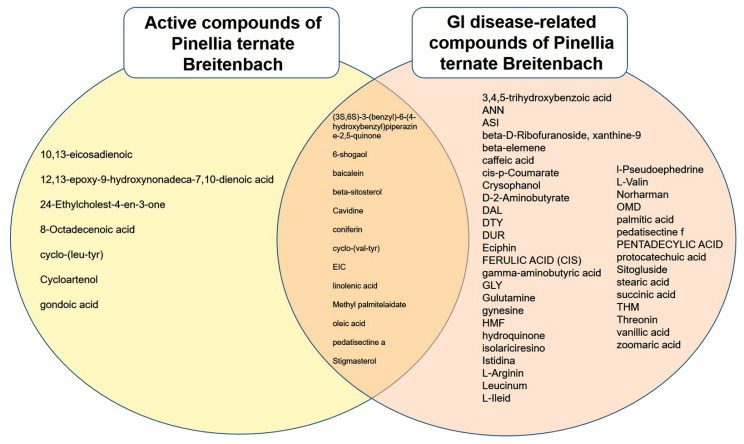
Venn diagram of the interactions between active compounds and gastrointestinal (GI) disease-related compounds in *Pinellia ternate Breitenbach*.

**Figure 4 plants-11-01348-f004:**
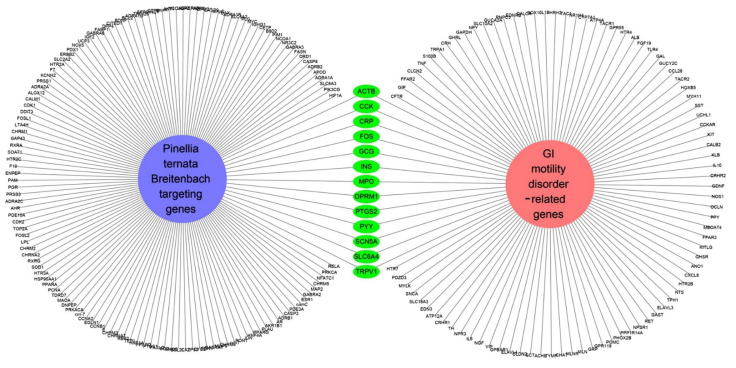
Network of gastrointestinal motility disorder-related genes and *Pinellia ternate Breitenbach*-targeting genes. The 13 genes included in both “genes related to gastrointestinal motility disorder” and “*Pinellia ternate Breitenbach*-target genes” are collected in the center.

**Figure 5 plants-11-01348-f005:**
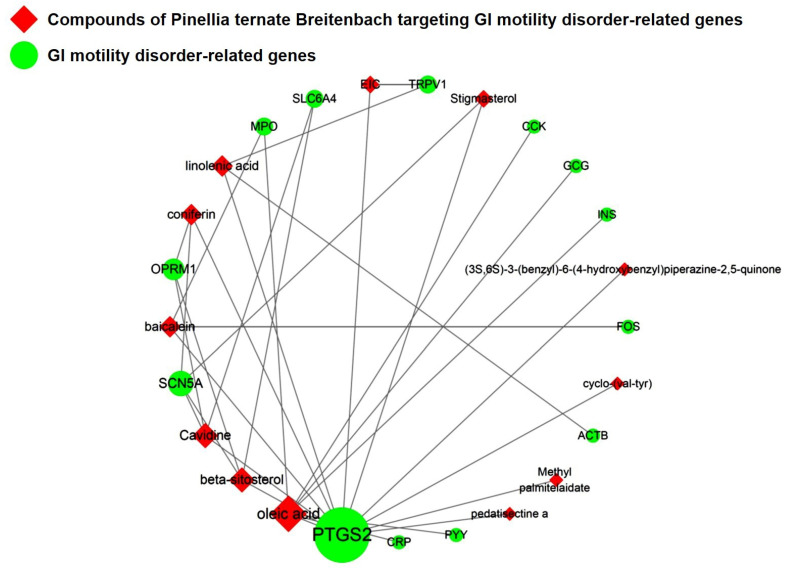
Network of compounds of Pinellia ternate Breitenbach and gastrointestinal (GI) motility disorder-related genes.

**Figure 6 plants-11-01348-f006:**
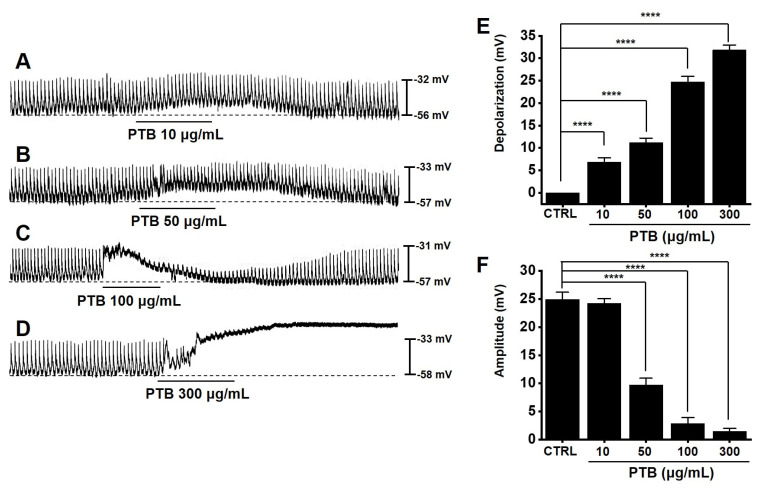
Effects of PTB extract on pacemaker potential of interstitial cells of Cajal (ICCs). (**A**–**D**) PTB extract depolarized the ICC pacemaker potentials. (**E**,**F**) The changes in pacemaker potential and amplitude induced by PTB extract are summarized. Means ± SEs. **** *p* < 0.0001. PTB: Pinellia ternate Breitenbach. CTRL: control.

**Figure 7 plants-11-01348-f007:**
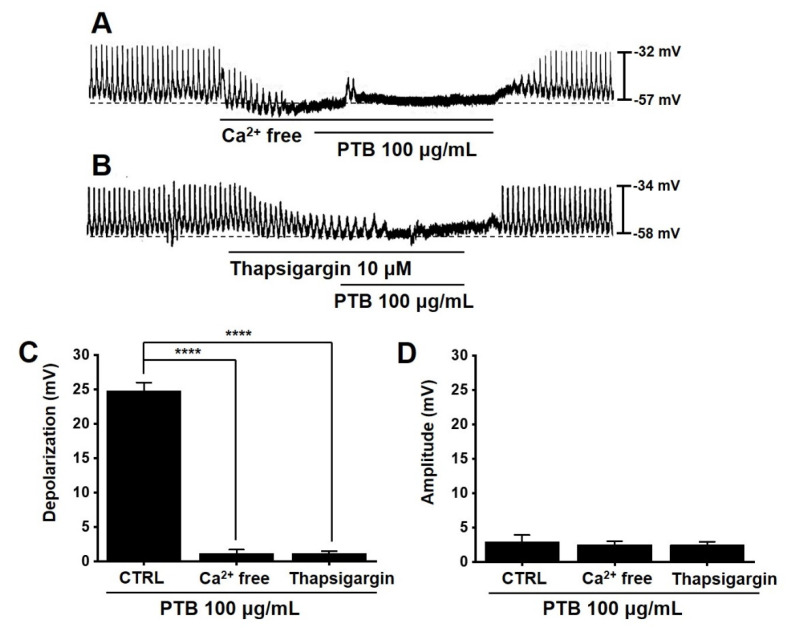
Effects of Ca^2+^ on PTB extract-induced pacemaker potential depolarization. (**A**) For external Ca^2+^-free solution, PTB extract did not result in depolarization. (**B**) With thapsigargin, PTB extract also did not result in depolarization. (**C**,**D**) Depolarization and amplitude responses to PTB extract are summarized. Means ± SEs. **** *p* < 0.0001. PTB: Pinellia ternate Breitenbach. CTRL: control.

**Figure 8 plants-11-01348-f008:**
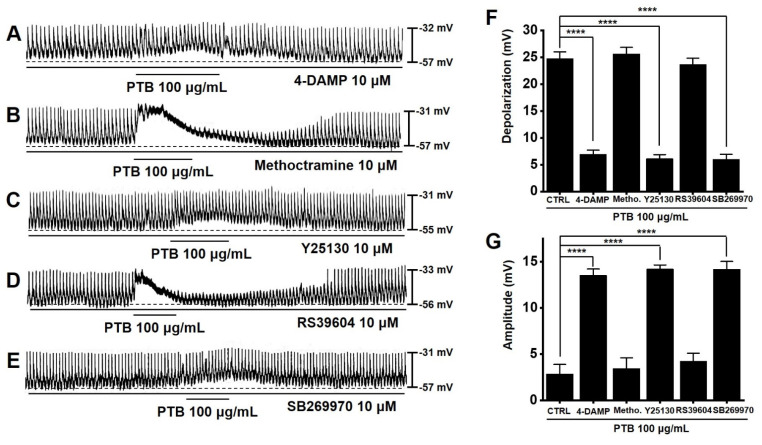
Effects of muscarinic and 5-HT receptor antagonists on PTB extract-induced pacemaker potential depolarization. (**A**) Pre-treatment with 4-DAMP inhibited PTB extract-induced effects. (**B**) Pre-treatment with methoctramine had no effects on PTB extract-induced effects. (**C**,**E**) Pre-treatment with Y25130 or SB269970 inhibited PTB extract-induced effects. (**D**) Pre-treatment with RS39604 had no effects on PTB extract-induced effects. (**F**,**G**) Depolarization and amplitude responses to PTB extract are summarized. Means ± SEs. **** *p* < 0.0001. PTB: Pinellia ternate Breitenbach. CTRL: control. Metho.: methoctramine.

**Figure 9 plants-11-01348-f009:**
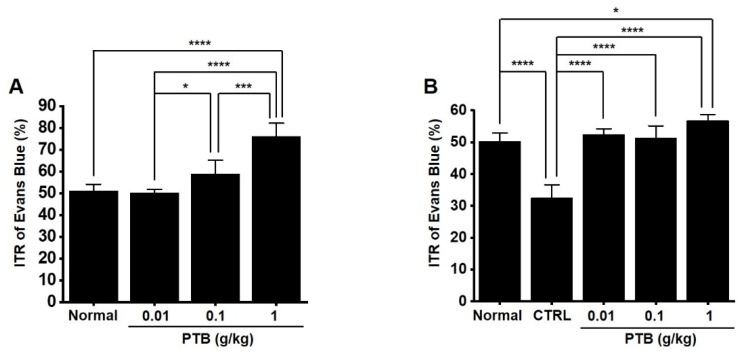
Effects of PTB extract on the intestinal transit rate (ITR) in normal and gastrointestinal motility disorder (GMD) mice. (**A**) PTB extract increased the ITR. (**B**) The ITR was recovered by PTB extract in GMD mice. Means ± SEs. * *p* < 0.05. *** *p* < 0.001. **** *p* < 0.0001. PTB: Pinellia ternate Breitenbach. CTRL: control.

**Table 1 plants-11-01348-t001:** Active compounds of Pinellia ternate Breitenbach.

Molecule Name	Structure	MW	OB (%) *	Caco-2 *	DL *
(3S,6S)-3-(benzyl)-6-(4-hydroxybenzyl)piperazine-2,5-quinone	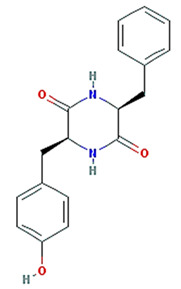	310.38	46.89	0.41	0.27
10,13-eicosadienoic	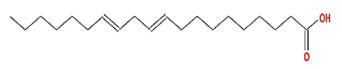	308.56	39.99	1.22	0.2
12,13-epoxy-9-hydroxynonadeca-7,10-dienoic acid	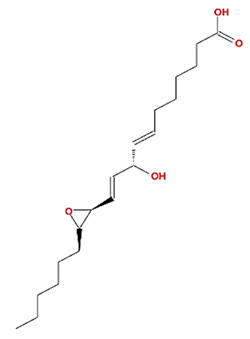	324.51	42.15	0.18	0.24
24-Ethylcholest-4-en-3-one	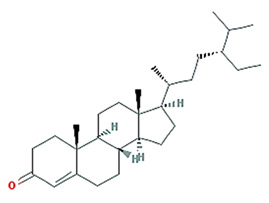	412.77	36.08	1.46	0.76
6-shogaol	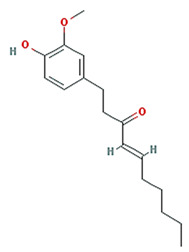	276.41	31	1.07	0.14
8-Octadecenoic acid	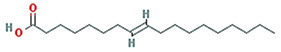	282.52	33.13	1.15	0.14
baicalein	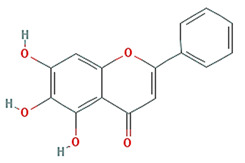	270.25	33.52	0.63	0.21
beta-sitosterol	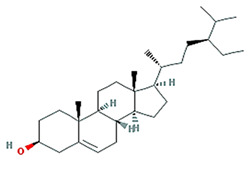	414.79	36.91	1.32	0.75
Cavidine	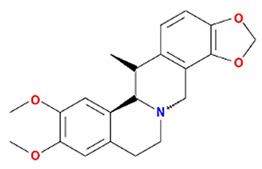	353.45	35.64	1.08	0.81
coniferin	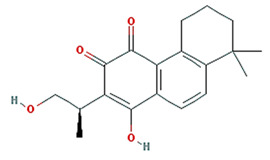	314.41	31.11	0.42	0.32
cyclo-(leu-tyr)	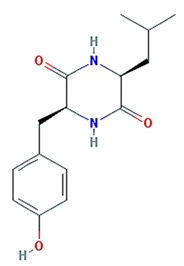	276.37	111.16	0.16	0.15
cyclo-(val-tyr)	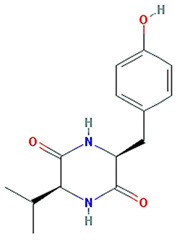	262.34	122.79	0.17	0.14
Cycloartenol	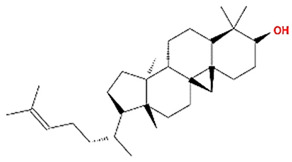	426.8	38.69	1.53	0.78
EIC	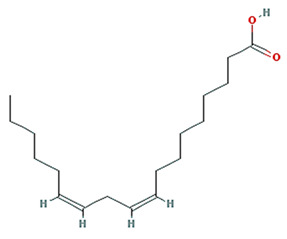	280.5	41.9	1.16	0.14
gondoic acid	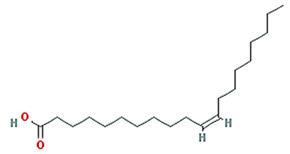	310.58	30.7	1.2	0.2
linolenic acid	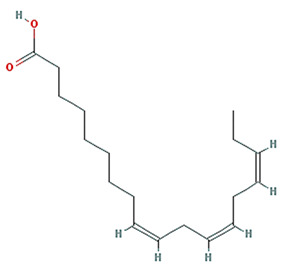	278.48	45.01	1.21	0.15
Methyl palmitelaidate	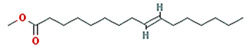	268.49	34.61	1.4	0.12
oleic acid	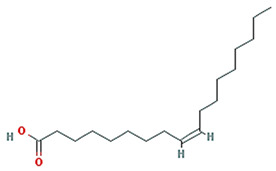	282.52	33.13	1.17	0.14
pedatisectine a	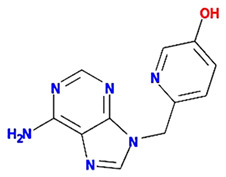	242.27	64.09	−0.3	0.16
Stigmasterol	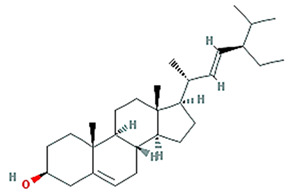	412.77	43.83	1.44	0.76

* OB: oral bioavailability; Caco-2: Caco-2 permeability; and DL: drug likeness.

**Table 2 plants-11-01348-t002:** Compounds and targets related to gastrointestinal (GI) disease.

Molecule Name	Gene Name	Disease Name
(3S,6S)-3-(benzyl)-6-(4-hydroxybenzyl)piperazine-2,5-quinone	PTGS2	GI motility disorder ^1^ Adenomatous polyposis Colorectal cancer Peutz–Jeghers syndrome
3,4,5-trihydroxybenzoic acid	HSP90AA1	Gastrointestinal stromal tumors (GIST)
	PTGS2	GI motility disorder ^1^ Adenomatous polyposis Colorectal cancer Peutz–Jeghers syndrome
6-shogaol	PPARG	Crohn’s disease, unspecified Inflammatory bowel disease Pancreatic cancer Ulcerative colitis
ANN	PTGS2	GI motility disorder ^1^ Adenomatous polyposis Colorectal cancer Peutz–Jeghers syndrome
ASI	ALOX5	Gastrointestinal cancers Inflammatory bowel disease Pancreatic cancer Ulcerative colitis
	NOS1	GI motility disorder ^1^
baicalein	FOS	GI motility disorder ^1^
	HSP90AA1	Gastrointestinal stromal tumors (GIST)
	MPO	GI motility disorder ^1^
	PTGS2	GI motility disorder ^1^ Adenomatous polyposis Colorectal cancer Peutz–Jeghers syndrome
beta-D-Ribofuranoside, xanthine-9	PTGS2	GI motility disorder ^1^ Adenomatous polyposis Colorectal cancer Peutz–Jeghers syndrome
beta-elemene	PTGS2	GI motility disorder ^1^ Adenomatous polyposis Colorectal cancer Peutz–Jeghers syndrome
beta-sitosterol	HSP90AA1	Gastrointestinal stromal tumors (GIST)
	OPRM1	GI motility disorder ^1^ Diarrhea
		Opioid-induced bowel dysfunction
	PTGS2	GI motility disorder ^1^ Adenomatous polyposis Colorectal cancer Peutz–Jeghers syndrome
	SCN5A	GI motility disorder ^1^
	SLC6A4	GI motility disorder ^1^
caffeic acid	PTGS2	GI motility disorder ^1^ Adenomatous polyposis Colorectal cancer Peutz–Jeghers syndrome
	TNF	GI motility disorder ^1^
		Crohn’s disease, unspecified
Cavidine	HSP90AA1	Gastrointestinal stromal tumors (GIST)
	HTR3A	Chemotherapy-induced nausea and vomiting Diarrhea Irritable bowel syndrome Postoperative nausea and vomiting
	OPRM1	GI motility disorder ^1^ Diarrhea Opioid-induced bowel dysfunction
	PTGS2	GI motility disorder ^1^ Adenomatous polyposis Colorectal cancer Peutz–Jeghers syndrome
	SCN5A	GI motility disorder ^1^
	SLC6A4	GI motility disorder ^1^
cis-p-Coumarate	PTGS2	GI motility disorder ^1^ Adenomatous polyposis Colorectal cancer Peutz–Jeghers syndrome
coniferin	CA2	Pancreatic cancer
	OPRM1	GI motility disorder ^1^ Diarrhea Opioid-induced bowel dysfunction
	PPARG	Crohn’s disease, unspecified Inflammatory bowel disease Pancreatic cancer Ulcerative colitis
	PTGS2	GI motility disorder ^1^ Adenomatous polyposis Colorectal cancer Peutz–Jeghers syndrome
	SCN5A	GI motility disorder ^1^
Crysophanol	HSP90AA1	Gastrointestinal stromal tumors (GIST)
	PTGS2	GI motility disorder ^1^ Adenomatous polyposis Colorectal cancer Peutz–Jeghers syndrome
	SCN5A	GI motility disorder ^1^
cyclo-(val-tyr)	PTGS2	GI motility disorder ^1^ Adenomatous polyposis Colorectal cancer Peutz–Jeghers syndrome
D-2-Aminobutyrate	NOS1	GI motility disorder ^1^
DAL	MMP12	Crohn’s disease, unspecified Gastrointestinal ulcers Ulcerative colitis
	NOS1	GI motility disorder ^1^
DTY	ACHE	GI motility disorder ^1^
	NOS3	Colon cancer
	PTGS2	GI motility disorder ^1^
		Adenomatous polyposis Colorectal cancer Peutz–Jeghers syndrome
DUR	CA2	Pancreatic cancer
	PTGS2	GI motility disorder ^1^ Adenomatous polyposis Colorectal cancer Peutz–Jeghers syndrome
Eciphin	ACHE	GI motility disorder ^1^
	NOS3	Colon cancer
	PTGS2	GI motility disorder ^1^ Adenomatous polyposis Colorectal cancer Peutz–Jeghers syndrome
	SCN5A	GI motility disorder ^1^
	SLC6A4	GI motility disorder ^1^
EIC	PTGS2	GI motility disorder ^1^ Adenomatous polyposis Colorectal cancer Peutz–Jeghers syndrome
	TRPV1	GI motility disorder ^1^
FERULIC ACID (CIS)	LTA4H	Oesophageal cancer
	NOS3	Colon cancer
	PTGS2	GI motility disorder ^1^ Adenomatous polyposis Colorectal cancer Peutz–Jeghers syndrome
gamma-aminobutyric acid	IL6	GI motility disorder ^1^
GLY	AMY2A	Pancreatic disease
	CTNNB1	Colorectal cancer
	LTA4H	Oesophageal cancer
	MMP12	Crohn’s disease, unspecified Gastrointestinal ulcers Ulcerative colitis
	NOS1	GI motility disorder ^1^
	PTGS2	GI motility disorder ^1^
		Adenomatous polyposis
		Colorectal cancer
		Peutz–Jeghers syndrome
Gulutamine	LTA4H	Oesophageal cancer
	NOS1	GI motility disorder ^1^
	PTGS2	GI motility disorder ^1^ Adenomatous polyposis Colorectal cancer Peutz–Jeghers syndrome
gynesine	PTGS2	GI motility disorder ^1^ Adenomatous polyposis Colorectal cancer Peutz–Jeghers syndrome
HMF	ACHE	GI motility disorder ^1^
	PTGS2	GI motility disorder ^1^ Adenomatous polyposis Colorectal cancer Peutz–Jeghers syndrome
hydroquinone	TNF	GI motility disorder ^1^
		Crohn’s disease, unspecified
isolariciresino	CA2	Pancreatic cancer
	HSP90AA1	Gastrointestinal stromal tumors (GIST)
	MAPK14	Crohn’s disease, unspecified
	NOS3	Colon cancer
	PPARG	Crohn’s disease, unspecified Inflammatory bowel disease Pancreatic cancer Ulcerative colitis
	PTGS2	GI motility disorder ^1^ Adenomatous polyposis Colorectal cancer Peutz–Jeghers syndrome
	SCN5A	GI motility disorder ^1^
Istidina	PTGS2	GI motility disorder ^1^ Adenomatous polyposis Colorectal cancer Peutz–Jeghers syndrome
L-Arginin	NOS1	GI motility disorder ^1^
	PTGS2	GI motility disorder ^1^ Adenomatous polyposis Colorectal cancer Peutz–Jeghers syndrome
Leucinum	NOS1	GI motility disorder ^1^
L-Ile	NOS1	GI motility disorder ^1^
linolenic acid	ACTB	GI motility disorder ^1^
	PTGS2	GI motility disorder ^1^ Adenomatous polyposis Colorectal cancer Peutz–Jeghers syndrome
	TRPV1	GI motility disorder ^1^
l-Pseudoephedrine	NOS3	Colon cancer
	PTGS2	GI motility disorder ^1^ Adenomatous polyposis Colorectal cancer Peutz–Jeghers syndrome
	SCN5A	GI motility disorder ^1^
	SLC6A4	GI motility disorder ^1^
L-Valin	NOS1	GI motility disorder ^1^
	PTGS2	GI motility disorder ^1^ Adenomatous polyposis Colorectal cancer Peutz–Jeghers syndrome
Methyl palmitelaidate	PTGS2	GI motility disorder ^1^ Adenomatous polyposis Colorectal cancer Peutz–Jeghers syndrome
Norharman	PTGS2	GI motility disorder ^1^ Adenomatous polyposis Colorectal cancer Peutz–Jeghers syndrome
oleic acid	CCK	GI motility disorder ^1^
	CRP	GI motility disorder ^1^
	GCG	GI motility disorder ^1^
	INS	GI motility disorder ^1^
	MPO	GI motility disorder ^1^
	PPARG	Crohn’s disease, unspecified Inflammatory bowel disease Pancreatic Cancer Ulcerative colitis
	PTGS2	GI motility disorder ^1^ Adenomatous polyposis Colorectal cancer Peutz–Jeghers syndrome
	PYY	GI motility disorder ^1^
OMD	PTGS2	GI motility disorder ^1^ Adenomatous polyposis Colorectal cancer Peutz–Jeghers syndrome
palmitic acid	IL10	GI motility disorder ^1^
	PTGS2	GI motility disorder ^1^ Adenomatous polyposis Colorectal cancer Peutz–Jeghers syndrome
	TNF	GI motility disorder ^1^
		Crohn’s disease, unspecified
pedatisectine a	PTGS2	GI motility disorder ^1^ Adenomatous polyposis Colorectal cancer Peutz–Jeghers syndrome
pedatisectine f	ACHE	GI motility disorder ^1^
	PTGS2	GI motility disorder ^1^ Adenomatous polyposis Colorectal cancer Peutz–Jeghers syndrome
PENTADECYLIC ACID	PTGS2	GI motility disorder ^1^ Adenomatous polyposis Colorectal cancer Peutz–Jeghers syndrome
protocatechuic acid	ALOX5	Gastrointestinal cancers Inflammatory bowel disease Pancreatic cancer
		Ulcerative colitis
	PTGS2	GI motility disorder ^1^ Adenomatous polyposis Colorectal cancer Peutz–Jeghers syndrome
Sitogluside	HSP90AA1	Gastrointestinal stromal tumors (GIST)
	HTR3A	Chemotherapy-induced nausea and vomiting Diarrhea Irritable bowel syndrome Postoperative nausea and vomiting
	PTGS2	GI motility disorder ^1^ Adenomatous polyposis Colorectal cancer Peutz–Jeghers syndrome
	SCN5A	GI motility disorder ^1^
stearic acid	PTGS2	GI motility disorder ^1^ Adenomatous polyposis Colorectal cancer Peutz–Jeghers syndrome
Stigmasterol	LTA4H	Oesophageal cancer
	PTGS2	GI motility disorder ^1^ Adenomatous polyposis Colorectal cancer Peutz–Jeghers syndrome
	SCN5A	GI motility disorder ^1^
succinic acid	NOS1	GI motility disorder ^1^
THM	CA2	Pancreatic cancer
	HSP90AA1	Gastrointestinal stromal tumors (GIST)
	PTGS2	GI motility disorder ^1^ Adenomatous polyposis Colorectal cancer Peutz–Jeghers syndrome
Threonin	NOS1	GI motility disorder ^1^
	PTGS2	GI motility disorder ^1^
		Adenomatous polyposis Colorectal cancer Peutz–Jeghers syndrome
vanillic acid	NOS3	Colon cancer
	PTGS2	GI motility disorder ^1^ Adenomatous polyposis Colorectal cancer Peutz–Jeghers syndrome
zoomaric acid	PTGS2	GI motility disorder ^1^ Adenomatous polyposis Colorectal cancer Peutz–Jeghers syndrome

^1^ After investigating the relationship between *Pinellia ternata Breitenbach* and GI motility disorder using Cytoscape stringApp, genes related to GI motility disorder were added to this table.

## Data Availability

The datasets used and/or analyzed during the current study are available from the corresponding author upon reasonable request.

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
