# Peer review of "Prediction of the Medicinal Mechanisms of Pinellia ternata Breitenbach, a Traditional Medicine for Gastrointestinal Motility Disorders, through Network Pharmacology"

_plants, 2022, doi:10.3390/plants11101348_

Round 1
Reviewer 1 Report
This article mentioned the action mechanism of PTB-extract that has not been clarified so far. Although each data is not relevant and the contents in this article are cross-the-board, the information obtained in this study seems to be important.
The experimental design and the results seem to be exquisite and valid. However, the following point would be considered to improve this article.
Minor point
1) Reviewer can understand that Ca2+ concentration in the cytoplasm is very important for the PTB-
extract action. What is the meaning that the responses of PBT-extract were same in two different
conditions, Ca2+-free condition (Ca2+ concentration in cytoplasm will be decreased) and
thapsigargin (Ca2+ concentration in cytoplasm will be increased) ?
2) Line 154, 163 and 164 : Ca2+ → Ca2+ (2+ ; superscript)
Line 221 : Na+ → Na+ (+ ; superscript)
Author Response
Reviewer1
This article mentioned the action mechanism of PTB-extract that has not been clarified so far. Although each data is not relevant and the contents in this article are cross-the-board, the information obtained in this study seems to be important.
The experimental design and the results seem to be exquisite and valid. However, the following point would be considered to improve this article.
Minor point
1) Reviewer can understand that Ca2+ concentration in the cytoplasm is very important for the PTB-extract action. What is the meaning that the responses of PBT-extract were same in two different conditions, Ca2+-free condition (Ca2+ concentration in cytoplasm will be decreased) and thapsigargin (Ca2+ concentration in cytoplasm will be increased)?
Responses) The Ca2+-free condition is when the concentration of extracellular calcium is 0 mM, and when thapsigargin is used, the function of SR, which is an intracellular calcium storage, is destroyed and the function of controlling the intracellular calcium concentration is suppressed. In this case, it is not known whether the intracellular calcium concentration will decrease or increase. However, it is thought that the intracellular calcium concentration will be decreased both when the extracellular calcium is 0 mM and when thapsigargin is used.
2) Line 154, 163 and 164 : Ca2+ → Ca2+ (2+ ; superscript)
Line 221 : Na+ → Na+ (+ ; superscript)
Responses) We changed.

Reviewer 2 Report
The reviewed manuscript presents a very interesting research project in which an evaluation of potential therapeutic targets for Pinellia ternata Breitenbach (PTB) in gastrointestinal motility disorders (GMDs) was performed using modern analytical tools (information platforms) before testing in laboratory animals. I view this project as novel and noteworthy. My minor comments relate to the presentation of the results and are for organizational purposes.
- Numerous abbreviations and acronyms are used in the text - they should be explained or expanded in the legend attached to the main text, or under the figures and tables;
- Tables 1 and 2 present PTB active ingredients in a disorderly way; they can be sorted by alphabetical order, by molecular weight, or by a chemical group; additionally, the names of compounds are written in different ways, once with a capital letter and once with a lowercase letter - which needs to be standardized
- The above remarks refer also to figures;
- The notation "beta" should be replaced with the appropriate Greek character;
- I recommend improving the readability of Figures 2, 4, and 5, especially the font size;
- The citation, i.e. the name of the author of the scientific name of the species in the title and the text should be written in normal font, not in italics.
Author Response
Reviewer2
The reviewed manuscript presents a very interesting research project in which an evaluation of potential therapeutic targets for Pinellia ternata Breitenbach (PTB) in gastrointestinal motility disorders (GMDs) was performed using modern analytical tools (information platforms) before testing in laboratory animals. I view this project as novel and noteworthy. My minor comments relate to the presentation of the results and are for organizational purposes.
- Numerous abbreviations and acronyms are used in the text - they should be explained or expanded in the legend attached to the main text, or under the figures and tables;
Responses) Overall, the contents of the use of abbreviations and acronyms were confirmed. Some are added to figures and tables, and some have confirmed that there is content in the text.
- Tables 1 and 2 present PTB active ingredients in a disorderly way; they can be sorted by alphabetical order, by molecular weight, or by a chemical group; additionally, the names of compounds are written in different ways, once with a capital letter and once with a lowercase letter - which needs to be standardized
Responses) We rearranged Tables 1 and 2 by alphabetical order. However, the name of compound was described based on the database of the original tcmsp. Therefore, it is judged that it is more correct to write the database as it is rather than standardize it.
- The above remarks refer also to figures;
Responses) The name of compound was described based on the database of the original tcmsp. Therefore, it is judged that it is more correct to write the database as it is rather than standardize it.
- The notation "beta" should be replaced with the appropriate Greek character;
Responses) The name of compound was described based on the database of the original tcmsp. Therefore, it is judged that it is more correct to write the beta (database) rather than the appropriate Greek character.
- I recommend improving the readability of Figures 2, 4, and 5, especially the font size;
Responses) We changed.
- The citation, i.e. the name of the author of the scientific name of the species in the title and the text should be written in normal font, not in italics.
Responses) We changed.